# A Real-World Experience from a Single Center (LPCE, Nice, France) Highlights the Urgent Need to Abandon Immunohistochemistry for ROS1 Rearrangement Screening of Advanced Non-Squamous Non-Small Cell Lung Cancer

**DOI:** 10.3390/jpm13050810

**Published:** 2023-05-10

**Authors:** Véronique Hofman, Samantha Goffinet, Christophe Bontoux, Elodie Long-Mira, Sandra Lassalle, Marius Ilié, Paul Hofman

**Affiliations:** 1Laboratory of Clinical and Experimental Pathology, University Côte d’Azur, FHU OncoAge, Pasteur Hospital, 06000 Nice, France; hofman.v@chu-nice.fr (V.H.); goffinet.s@chu-nice.fr (S.G.); bontoux.c@chu-nice.fr (C.B.); long-mira.e@chu-nice.fr (E.L.-M.); lassalle.s@chu-nice.fr (S.L.); ilie.m@chu-nice.fr (M.I.); 2Hospital-Integrated Biobank (BB-0033-00025), Pasteur Hospital, 06000 Nice, France; 3Team 4, IRCAN Inserm U1081, CNRS 7284, Université Côte d’Azur, 06100 Nice, France

**Keywords:** *ROS1* rearrangement, next generation sequencing, immunohistochemistry, fluorescent in situ hybridization, algorithm

## Abstract

The detection of *ROS1* rearrangements in metastatic non-squamous non-small cell lung carcinoma (NS-NSCLC) permits administration of efficient targeted therapy. Detection is based on a testing algorithm associated with ROS1 immunohistochemistry (IHC) screening followed by *ROS1* FISH and/or next generation sequencing (NGS) to confirm positivity. However, (i) *ROS1* rearrangements are rare (1–2% of NS-NSCLC), (ii) the specificity of ROS1 IHC is not optimal, and (iii) *ROS1* FISH is not widely available, making this algorithm challenging to interpret time-consuming. We evaluated RNA NGS, which was used as reflex testing for *ROS1* rearrangements in NS-NSCLC with the aim of replacing ROS1 IHC as a screening method. ROS1 IHC and RNA NGS were prospectively performed in 810 NS-NSCLC. Positive results were analyzed by *ROS1* FISH. ROS1 IHC was positive in 36/810 (4.4%) cases that showed variable staining intensity while NGS detected *ROS1* rearrangements in 16/810 (1.9%) cases. *ROS1* FISH was positive in 15/810 (1.8%) of ROS1 IHC positive cases and in all positive ROS1 NGS cases. Obtaining both ROS1 IHC and *ROS1* FISH reports took an average of 6 days, while obtaining ROS1 IHC and RNA NGS reports took an average of 3 days. These results showed that systematic screening for the ROS1 status using IHC must be replaced by NGS reflex testing.

## 1. Introduction

The detection of *ROS1* rearrangement in stage IIIB/IV non-small cell lung cancer (NSCLC) patients makes them eligible for targeted therapy, notably for administration of the tyrosine-kinase inhibitor crizotinib [1,2]. Therefore, it is necessary to systematically look for *ROS1* rearrangements in cases of metastatic non-squamous NSCLC (NS-NSCLC) at the baseline [3,4]. Most of the algorithms for *ROS1* status assessment in routine clinical practice, notably in pathology laboratories, are based on ROS1 immunohistochemistry (IHC) as a screening test, followed by a molecular biology test [*ROS1* FISH, next generation sequencing (NGS), or RT-qPCR approaches] to confirm ROS1 IHC positivity [3,5,6,7,8,9,10,11,12]. It is noteworthy that *ROS1* targeted therapy can be administered only based on a positive molecular biology result. Therefore, this two-step algorithm is necessary considering the variable specificity and the high sensitivity of ROS1 IHC staining, which may give false positive results and, less frequently, false negative results [5]. 

The progressive discovery of several therapeutic molecular targets for NS-NSCLC has increased the number of biomarkers to analyze. Thus, it is recommended, more and more often, to evaluate not only the *EGFR*, *ALK,* and *ROS1* status, but also that of *BRAF, RET*, *NTRK*, *MET*, *KRAS,* and *HER2* before any treatment [4]. Therefore, it is difficult in daily practice to sequentially look for the genomic alterations of each of these genes for several reasons. First, the turnaround times (TAT) to obtain most of the results can delay the initiation of targeted treatment. Second, some of these tests cannot be performed or lead to uncertain or false negative results due to small tissue biopsies and/or a low percentage of tumor cells, which give an insufficient amount of extracted nucleic acid and/or tumor cells visible on tissue sections [13,14]. Consequently, at present, the need to screen for ROS1 with IHC is under discussion, and its use is being challenged. This paves the way for DNA and RNA NGS reflex testing, which, in a single step, can look at all the necessary genomic alterations associated with the currently available targeted therapies used in routine clinical practice [4]. 

This study aimed to compare, prospectively, the results of *ROS1* rearrangement screening by ROS1 IHC and RNA NGS for 810 NS-NSCLC patients. The results were then compared to those of the *ROS1* FISH. Finally, the TAT to obtain the different results were also compared.

## 2. Patients and Methods

Between September 2021 and February 2023, 810 NS-NSCLC patients were tested systematically by ROS1 IHC and DNA and RNA NGS (Laboratory of Clinical and Experimental Pathology, Pasteur Hospital, University Côte d’Azur Nice, France; Figure 1). 

The main epidemiological, clinical, and pathological data are shown in Table 1. 

ROS1 IHC and/or ROS1 NGS positive cases were analyzed with *ROS1* FISH. The TAT was considered to be the time between obtaining the histological diagnosis and electronic validation of the reports.

### 2.1. ROS1 Immunohistochemistry

The formalin-fixed, paraffin-embedded (FFPE) tissue, with a thickness of 4 μm, was subjected to immunohistochemistry. A rabbit monoclonal ROS1 antibody (D4D6) provided by Cell Signaling Technology (Danvers, MA, USA) was used at a 1:75 dilution for a duration of 2 hours. The Benchmark ULTRA autostainer (Ventanaa, Tucson, AZ, USA) was used to perform staining, which was achieved through the OptiView DAB IHC detection kit (Roche) and an amplification kit (Ventana). To evaluate the staining intensity for each case, an assessment was conducted. An H-score, ranging from 0 to 300, was calculated for each case by multiplying the staining intensity (0 indicating no staining, 1 representing weak staining but more than the background staining, 2 indicating moderate staining, and 3 indicating strong staining) by the percentage of positive tumor cells. The positive controls included ROS1-rearranged lung adenocarcinoma tissue that tested FISH positive.

### 2.2. Next Generation Sequencing

The nucleic acid extraction was conducted using either the Maxwell RSC Instrument (Promega, Madison, WI, USA) in combination with the Maxwell RSC FFPE Plus DNA kit or the Maxwell RSC RNA FFPE kit (Promega). After the extraction of nucleic acid, the concentration was measured by employing the Qubit Fluorometric quantification assay (Thermo Fisher Scientific, Waltham, MA, USA), utilizing the Qubit RNA HS Assay Kit and the Qubit dsDNA HS Assay Kit. The Ion Torrent™ Genexus™ Integrated Sequencer (Thermo Fisher Scientific) was used for the detection of genomic alterations by Ion semiconductor sequencing (Ion Torrent™ Technology, Thermo Fisher Scientific). The Oncomine™ Precision Assay GX panel (OPA) provided by Thermo Fisher Scientific was utilized, targeting 50 key genes. Among them, 45 genes were designed for DNA mutation detection, 18 for fusion detection, and 14 for copy number variant (CNV) detection. In addition, the panel incorporated a 5′/3′ expression imbalance strategy for the detection of novel fusions. By using this panel, up to 16 samples could be sequenced simultaneously on a single run with the Genexus sequencer. 

### 2.3. ROS1 Fluorescence In Situ Hybridization

De-paraffinization of 4-micron formalin-fixed, paraffin-embedded tissue sections was performed before conducting a pre-treatment step of heat-induced epitope retrieval (HIER) using SPoT-Light Tissue Pretreatment Solution (Thermo Fisher Scientific) at pH 7. A proteolytic digestion of tissue sections was then carried out using Protease 1 (Abbott Molecular, Des Plaines, IL, USA), followed by a rinse in saline-sodium citrate buffer (SSC). The ZytoLight SPEC ROS1 Dual Colour Break Apart Probe (ZytoVision, Bremerhaven, Germany) was used, followed by denaturation at 95 °C for 5 min and overnight hybridization at 37 °C. The slides were then dehydrated and counterstained with SlowFade Gold DAPI (Invitrogen, Waltham, MA, USA). At least 50 tumor nuclei per case were evaluated for interphase signals using an epifluorescence microscope (Zeiss, White Plains, NY, USA) and an automated fluorescence microscope (Olympus, Tokyo, Japan) equipped with cell imaging and analysis software (BioView, Rehovot, Israel). Cases were categorized as ROS1 FISH positive if they demonstrated at least 15% of cells with split signals at least one signal distance apart or an isolated centromeric 30 (green signal) pattern. This 15% cutoff was determined by in-house validation and adheres to international guidelines [15].

## 3. Results

ROS1 IHC was scored in 810 cases of NS-NSCLC. Detectable staining was seen in 36/810 (4.4%) of cases showing a variable intensity and giving an H-score (Figure 2; Table 2). 

*ROS1* FISH was positive in 15 of the 36 cases (42%, Figure 3). 

RNA NGS detected *ROS1* rearrangement in 16 of the 36 (44%) ROS1 IHC positive cases and included one negative case by *ROS1* FISH (Table 2). Two cases were excluded as false positive as an EML4-*ALK* fusion was detected by RNA sequencing, and no *ROS1* structural variants were seen despite good coverage of all *ROS1* introns and exons. These cases were positive for ALK IHC and *ALK* FISH and had an H-score of 100 for ROS1. None of the negative ROS1 IHC was found to be positive for *ROS1* rearrangement when using RNA NGS.

The TAT was 4 days (range 3–8 days) to obtain DNA and RNA NGS reports. The TAT was, in total, 5 days (range 4–9 days) to obtain both the ROS1 IHC (2 days; range 2–4 days) and the *ROS1* FISH (3 days; range 2–5 days) reports. All driver mutations were mutually exclusive. 

## 4. Discussion

This study showed that an ultrafast DNA and RNA NGS setting at the baseline can be greatly involved in reflex testing for *ROS1* rearrangements in NS-NSCLC [16,17]. Here, the NGS results were obtained in an average of four working days and a positive result was confirmed in 15/16 cases by FISH. Therefore, one false negative *ROS1* FISH was observed due to the presence of short deletion, as previously reported [18]. In comparison, the ROS1 IHC results were obtained in an average of two working days, but the positivity was only confirmed after an additional three working days by *ROS1* FISH in 42% of the cases, highlighting the relatively weak specificity of ROS1 IHC. Moreover, as already reported, false positive ROS1 IHC cases due to the presence of an *ALK* rearrangement were observed in our series [19].

Taken together, this led us to abandon the ROS1 IHC screening approach and the sequential ROS1 IHC/*ROS1* FISH algorithm in our laboratory. Thus, a new algorithm was set up to evaluate the ROS1 status by first systematically performing ROS1 RNA NGS and then confirming the positive results with *ROS1* FISH. Globally, the percentage of positive cases of *ROS1* rearrangement was a little bit lower than that described in the literature since our cohort of NS-NSCLC patients contains more than 30% of patients with early-stage disease, which is associated with a lower number of rearranged *ROS1* tumors [20].

The shift in routine clinical practice away from ROS1 IHC screening of patients’ NS-NSCLC was warranted for different reasons, notably based on the results obtained in the present work. ROS1 status evaluation using IHC has a couple of limitations. It has been shown to give a number of false positive results, and the low specificity has been confirmed by discrepancies with the *ROS1* FISH results [5]. Moreover, it is important to consider which ROS1 antibody is being used since the specificity and sensitivity of these antibodies can be variable [21,22,23,24]. The determination of an H-score from ROS1 IHC can have added value to set up a cutoff for whether to perform *ROS1* FISH to confirm the results [25]. This was not confirmed in the present study. One reason for this could be the low number of positive ROS1 cases detected in our series. Therefore, in contrast to ALK IHC, which can be used as a companion diagnostic test, ROS1 IHC always has to be confirmed by at least one molecular biology test [26,27]. The relatively low frequency of *ROS1* rearrangements in NS-NSCLC, present in approximatively 0.9–2.6% according to the series, which leads to a ROS1 IHC for a low number of positive results, needs to be taken into consideration [28]. Moreover, more rarely, some false negative results have been noted with ROS1 IHC [7,8,29]. Finally, for many cases, both ROS1 IHC and *ROS1* FISH are unnecessarily performed, resulting in associated costs, an increase in the work load in the pathology laboratory, and an increase in the TAT to obtain reports; this can also lead to exhaustion of the tumor tissue, notably in cases of small biopsies and/or low percentages of tumor cells [13,14]. 

Our results demonstrated perfect concordance between the *ROS1* FISH results and those obtained with RNA NGS. RNA NGS was combined simultaneously with DNA NGS, which allowed assessment of the status of the current necessary genes at the baseline for NS-NSCLC [4]. One previous limitation to the use of only NGS for *ROS1* rearrangement evaluation was the delay in obtaining the results, which would not be suitable for the administration of a targeted therapy according to the organization, workflow of the samples, and sequencing approaches. However, the development of a new ultrafast NGS system has allowed us to obtain, in another study, the results in an average of four days for positive cases, which is even faster than the TAT when sequentially using ROS1 IHC and then *ROS1* FISH [16,17]. It is also noteworthy that RNA NGS can lead to the identification of all the fusion partners of *ROS1* and allow the detection of short deletions sometimes not visible with *ROS1* FISH [30,31]. Although DNA-based sequencing can detect rearrangements of fusion genes in intron regions, those genes often differ from messenger RNA (mRNA) fusions, emphasizing that RNA NGS is certainly the ideal approach for the evaluation of the *ROS1* status [30,32,33]. Performing DNA/RNA NGS at the same time allows for the investigation of the different genomic alterations that are currently necessary to evaluate at the baseline in NS-NSCLC [4]. In this context, it is noteworthy that concurrent classic driver oncogene mutations with *ROS1* rearrangement may predict a superior clinical outcome in NSCLC patients [34]. However, a number of restrictions can limit the possibility of developing, in daily practice, the new algorithm described above. First, in contrast to immunohistochemical platforms that are largely available in the majority of pathology laboratories, NGS approaches are not equally distributed in all countries or even in organizations and institutions in a single country [35,36]. Second, even if available, access to some NGS approaches could be limited, due to the cost and absence of reimbursement of costs, and can also be associated with a long TAT to obtain the results, which is not compatible with international guidelines [15]. Due to the lower cost and the shorter TAT, IHC and/or rapid RT-PCR can be easily performed as an alternative for ROS1 status assessment [16,36,37,38]. More importantly, IHC and FISH methods can sometimes be the only approaches that detect some therapeutic targets in very small tissue biopsies and/or, if only a low percentage of tumor cells are present, knowing that NGS can lead to some false negative results due to the low quantity and/or quality of the extracted nucleic acid [39,40]. As mentioned previously, in contrast to RNA NGS, *ROS1* FISH can also lead to false negative results, notably in the presence of small deletions [41]. Moreover, *ROS1* FISH has a relatively high price tag and technical difficulties (a sufficient number of tumor cells are needed) and can be time-consuming for the operator. 

Thus, it seems important to integrate different approaches for *ROS1* status evaluation into a pathology laboratory. *ROS1* FISH is useful, notably as an orthogonal tool, in addition to ROS1 IHC to validate some uncertain NGS and/or RT-PCR results, notably in young and non-smoker patients. In addition, new *in situ* technologies, such as multiplex IHC, could be associated with different antibodies, including a ROS1 antibody. This approach can not only save tissue material, but it can also reduce the TAT to obtain, at the same time, results associated with different targetable molecules [42].

## 5. Conclusions

In conclusion, the present study demonstrated that performing IHC to evaluate the *ROS1* rearrangement status in advanced NS-NSCLC should be abandoned nowadays in favor of ultrafast RNA NGS reflex testing [16,17,43]. *ROS1* FISH is still useful for validation of the diagnosis in the case of uncertain NGS results and/or of very small tissue biopsies with a few tumor cells. Finally, *ROS1* FISH can also be performed if RNA NGS is negative under certain circumstances (young and/or non-smoker patients, or specific requests by physicians). 

## Figures and Tables

**Figure 1 jpm-13-00810-f001:**
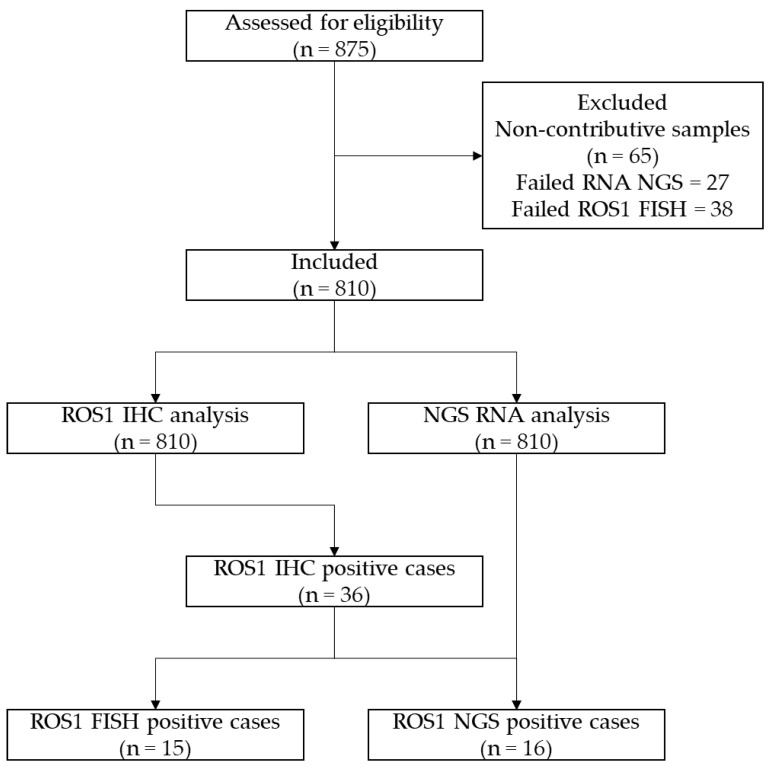
CONSORT flow diagram of the study.

**Figure 2 jpm-13-00810-f002:**
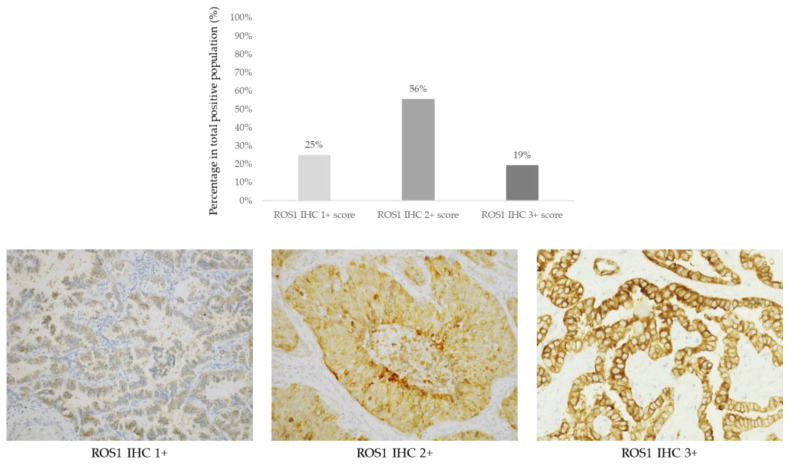
**Upper panel**: Prevalence of ROS1 positive cases (n = 36) according to IHC scores 1+ (9/36, 25%), 2+ (20/36, 56%), and 3+ (7/36, 19%). **Lower panel**: Representative images of non-squamous non-small cell lung cancer cases according to ROS1 IHC scores (D4D6 clone, immunoperoxidase, original magnification ×200).

**Figure 3 jpm-13-00810-f003:**
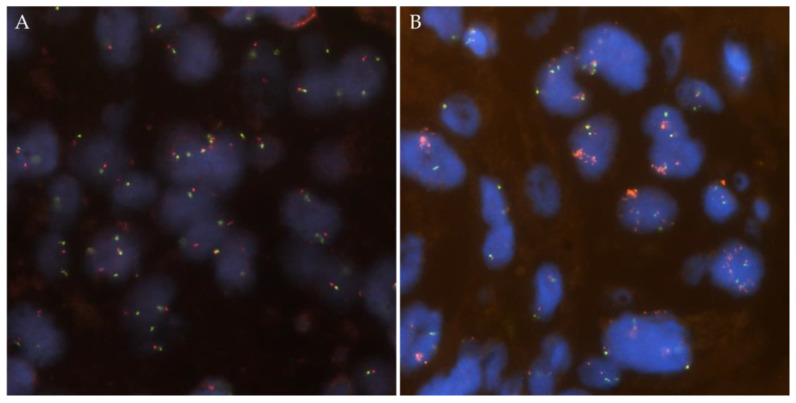
Representative images from *ROS1* copy number alterations detected by FISH. (**A**) *ROS1* wild-type case with two green/orange fusion signals per tumor cell. (**B**) ROS1 amplified case with *ROS1* cluster amplification in tumor nuclei with two green probes.

**Table 1 jpm-13-00810-t001:** Main epidemiological and pathological data.

Variable	Type	n (%)
All patients (n = 810)		
Age [median (range)]	62 (44–81) years old	
Gender		
	Male	535 (66%)
	Female	275 (24%)
Smoking status	Smoker	577 (71%)
	Non-smoker	71 (9%)
	Former smoker	162 (20%)
Histological subtypes	Acinar adenocarcinoma	450 (55%)
	Micropapillary adenocarcinoma	35 (4%)
	Mucinous invasive adenocarcinoma	40 (5%)
	Papillary adenocarcinoma	120 (15%)
	Solid adenocarcinoma	47 (6%)
	Non-small cell carcinoma, NOS	38 (5%)
	Non-small cell carcinoma, favoring adenocarcinoma	80 (10%)
pTNM stage	III	190 (23%)
	IV	620 (77%)
ROS1 H-score (positive cases)	Mean (standard deviation)	110 (30–300)
	Median (range)	120 (40–300)

**Table 2 jpm-13-00810-t002:** Number of ROS1 positive cases according to the testing methods.

ROS1 Testing Method	Number of Positive Cases	Percentage of Positive Cases (Total = 810)
IHC	36	4.44%
FISH	15	1.85%
RNA NGS	16	1.97%

Note: Two cases were excluded due to false positivity, and they were positive for *ALK* IHC and *ALK* FISH.

## Data Availability

No new data were created or analyzed in this study. Data sharing is not applicable to this article.

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
