# Peer review of "A Real-World Experience from a Single Center (LPCE, Nice, France) Highlights the Urgent Need to Abandon Immunohistochemistry for ROS1 Rearrangement Screening of Advanced Non-Squamous Non-Small Cell Lung Cancer"

_jpm, 2023, doi:10.3390/jpm13050810_

Round 1
Reviewer 1 Report
Hofman and colleagues have written an important paper on optimal testing strategies for ROS1-rearrangement in non-squamous NSCLC. While I agree with the message of the manuscript, I believe a number of changes are necessary before it can be published.
First, the authors should give a clearer indication of the patient flow in their study, with more detail about the exact strategies being compared. A CONSORT diagram would be ideal but a paragraph in the results should suffice. Specifically, the reader should be able to know:
1) How many patients received each test. Were all 810 patients tested with IHC and NGS? Did all patients undergo FISH testing, or was this restricted to IHC-positive / NGS-positive cases? Having read the manuscript a few times, I'm still not 100% sure.
2) How many patients received an indeterminate result with each method.
Second, I believe their comparison of TAT is not entirely accurate.
1) On lines 192-193, the authors claim that the NGS strategy is faster than IHC+FISH. However, if the competing algorithms are IHC+FISH versus NGS alone, NGS is only faster at identifying positive cases (negative cases are identified in 2 days with IHC versus 4 days with NGS). Given that most patients are ROS1-negative, I would assume that testing with NGS is slower in general.
2) If the competing algorithms are IHC+FISH versus NGS+FISH, NGS+FISH is not faster than IHC+FISH at all (2+3 days with IHC+FISH is shorter than 4+3 days with NGS+FISH).
3) Comparisons of TAT should be supported by some kind of statistical analysis. I have attached some R code (performing a paired Wilcoxon signed rank test) that the authors may use. I've simulated some data at the top which should be substituted with real data.
As I see it, the time advantage with NGS is the ability to test for multiple oncogenes simultaneously. But when looking at ROS1 alone, the IHC+FISH algorithm is faster. If the authors wish to make an argument in favour of NGS on the basis of timeliness, they can say that all samples were tested within a reasonable timeframe - possibly with reference to local/institutional guidelines for timely testing (e.g., 1 week).
Minor comments and suggestions:
- Page 5, lines 143-145: Placeholder text
- The numbers in the abstract do not match the results in the text.
- There is no data analysis section in the methods. What software did the authors use?
- The authors mention DNA and RNA NGS testing in the methods and should clarify which method was used for what purpose. Was RNA NGS used to test for ROS1 in all cases, or was DNA NGS used in some instances? If the latter, table 2 should be amended.
Author Response
Authors’ responses to reviewers
Reviewer #1
Hofman and colleagues have written an important paper on optimal testing strategies for ROS1-rearrangement in non-squamous NSCLC. While I agree with the message of the manuscript, I believe a number of changes are necessary before it can be published.
Authors’ response: We thank the reviewer for her/his favorable evaluation on our study. Thank you for your important suggestions that helped us to improve the manuscript.
First, the authors should give a clearer indication of the patient flow in their study, with more detail about the exact strategies being compared. A CONSORT diagram would be ideal but a paragraph in the results should suffice. Specifically, the reader should be able to know:
1) How many patients received each test. Were all 810 patients tested with IHC and NGS? Did all patients undergo FISH testing, or was this restricted to IHC-positive / NGS-positive cases? Having read the manuscript a few times, I'm still not 100% sure.
Authors’ response: Thank you very much for this comment. Indeed, all 810 patients were tested with IHC and NGS (page 2, line 64). All positive cases underwent FISH testing (page 3, line 71).
We have now showed the CONSORT Flow Diagram of the study in Figure 1.
2) How many patients received an indeterminate result with each method.
Authors’ response: Thank you for this important comment. We have now showed the non-contributive results in the CONSORT Flow Diagram. These cases were excluded for the final analysis of the study.
Second, I believe their comparison of TAT is not entirely accurate.
1) On lines 192-193, the authors claim that the NGS strategy is faster than IHC+FISH. However, if the competing algorithms are IHC+FISH versus NGS alone, NGS is only faster at identifying positive cases (negative cases are identified in 2 days with IHC versus 4 days with NGS). Given that most patients are ROS1-negative, I would assume that testing with NGS is slower in general.
Authors’ response: Thank you for your valuable comment. We totally agree with the reviewer that NGS is only faster at identifying positive cases. We have now modified the discussion section accordingly.
2) If the competing algorithms are IHC+FISH versus NGS+FISH, NGS+FISH is not faster than IHC+FISH at all (2+3 days with IHC+FISH is shorter than 4+3 days with NGS+FISH).
Authors’ response: Thank you for this important comment. We have rephrased the sentence in the discussion section. In our experience, the “standard” algorithm IHC+FISH in house is longer (2+5 days) for positive cases versus the new ultrafast NGS system (4 days), used in house as reflex-testing without confirmatory FISH analysis, as we showed in recent studies (PMID: 36718140; PMID: 3556538).
3) Comparisons of TAT should be supported by some kind of statistical analysis. I have attached some R code (performing a paired Wilcoxon signed rank test) that the authors may use. I've simulated some data at the top which should be substituted with real data.
As I see it, the time advantage with NGS is the ability to test for multiple oncogenes simultaneously. But when looking at ROS1 alone, the IHC+FISH algorithm is faster. If the authors wish to make an argument in favour of NGS on the basis of timeliness, they can say that all samples were tested within a reasonable timeframe - possibly with reference to local/institutional guidelines for timely testing (e.g., 1 week).
Authors’ response: Thank you for your valuable comment. However, as discussed and showed above, in our experience and based on the results of this study as well as recent published results from laboratory, the ultra-fast NGS approach is faster than other testing strategy/algorithm.
Minor comments and suggestions:
- Page 5, lines 143-145: Placeholder text
Authors’ response: We corrected the Placeholder text.
- The numbers in the abstract do not match the results in the text.
Authors’ response: Thank you very much. We now corrected the numbers in the abstract.
- There is no data analysis section in the methods. What software did the authors use?
Authors’ response: Given the results, there was no use for statistical analysis, except for Excel.
- The authors mention DNA and RNA NGS testing in the methods and should clarify which method was used for what purpose. Was RNA NGS used to test for ROS1 in all cases, or was DNA NGS used in some instances? If the latter, table 2 should be amended.
Authors’ response: All cases were analyzed by NGS RNA and NGS DNA assays in parallel, according to our laboratory’s standard operation procedures. However, it seems for us than the NGS DNA testing is out of the scope of the present work and did not included then the obtained data.
Reviewer 2 Report
Dear author, I am writing about your paper A real-world experience from a single center (LPCE, Nice, France) highlights the urgent need to abandon immunohistochemistry for ROS1 rearrangement screening of advanced non-squamous non-small cell lung cancer. The overall scientific idea behind this paper is very interesting and exciting. But I have some concerns.
Where is the data from NSG and FISH in the main file? There is only ROS1 immunohistochemistry data.
Author Response
Dear author, I am writing about your paper A real-world experience from a single center (LPCE, Nice, France) highlights the urgent need to abandon immunohistochemistry for ROS1 rearrangement screening of advanced non-squamous non-small cell lung cancer. The overall scientific idea behind this paper is very interesting and exciting. But I have some concerns.
Where is the data from NSG and FISH in the main file? There is only ROS1 immunohistochemistry data.
Authors’ response: Thank you for your valuable comment. We appreciate the reviewer’s input. We now have included in the revised Manuscript the positive results of all assays in the recently added CONSORT Flow Diagram. In addition, Table 2 shows the positive rate of the study. Furthermore, we have provided a comparison of the turnaround times for both methods. We have also included an explanation for the few discordant cases that were observed between all methods.
Round 2
Reviewer 2 Report
Dear author thanks for your response. It would be great if you provide FISH images in the main file as figure.
As per my knowledge any imaging experiment is incomplete without showing any representative images. only statistical representation is not sufficient for publishing a good paper.
Author Response
Thank you for this important comment. We have now added a new Figure 3 showing FISH ROS1 images from representative cases.
Round 3
Reviewer 2 Report
Thanks for your reply.